# Group B *Streptococcus* CRISPR1 Typing of Maternal, Fetal, and Neonatal Infectious Disease Isolates Highlights the Importance of CC1 in *In Utero* Fetal Death

Brice Le Gallou,[a,b] Adeline Pastuszka,[a,b] Coralie Lemaire,[a,b] Laurent Mereghetti,[a,b] Philippe Lanotte[a,b]

aUniversité de Tours, INRAE, Infectiologie et Santé Publique, BRMF, Tours, France
bService de Bactériologie-Virologie, Centre Hospitalier Régional Universitaire de Tours, Tours, France

**ABSTRACT** We performed a descriptive analysis of group B *Streptococcus* (GBS) isolates responsible for maternal and fetal infectious diseases from 2004 to 2020 at the University Hospital of Tours, France. This represents 115 isolates, including 35 isolates responsible for early-onset disease (EOD), 48 isolates responsible for late-onset disease (LOD), and 32 isolates from maternal infections. Among the 32 isolates associated with maternal infection, 9 were isolated in the context of chorioamnionitis associated with *in utero* fetal death. Analysis of neonatal infection distribution over time highlighted the decrease in EOD since the early 2000s, while LOD incidence has remained relatively stable. All GBS isolates were analyzed by sequencing their CRISPR1 locus, which is an efficient way to determine the phylogenetic affiliation of strains, as it correlates with the lineages defined by multilocus sequence typing (MLST). Thus, the CRISPR1 typing method allowed us to assign a clonal complex (CC) to all isolates; among these isolates, CC17 was predominant (60/115, 52%), and the other main CCs, such as CC1 (19/115, 17%), CC10 (9/115, 8%), CC19 (8/115, 7%), and CC23 (15/115, 13%), were also identified. As expected, CC17 isolates (39/48, 81.3%) represented the majority of LOD isolates. Unexpectedly, we found mainly CC1 isolates (6/9) and no CC17 isolates that were responsible for *in utero* fetal death. Such a result highlights the possibility of a particular role of this CC in *in utero* infection, and further investigations should be conducted on a larger group of GBS isolated in a context of *in utero* fetal death.

**IMPORTANCE** Group B *Streptococcus* is the leading bacterium responsible for maternal and neonatal infections worldwide, also involved in preterm birth, stillbirth, and fetal death. In this study, we determined the clonal complex of all GBS isolates responsible for neonatal diseases (early- and late-onset diseases) and maternal invasive infections, including chorioamnionitis associated with *in utero* fetal death. All GBS was isolated at the University Hospital of Tours from 2004 to 2020. We described the local group B *Streptococcus* epidemiology, which confirmed national and international data concerning neonatal disease incidence and clonal complex distribution. Indeed, neonatal diseases are mainly characterized by CC17 isolates, especially in late-onset disease. Interestingly, we identified mainly CC1 isolates responsible for *in utero* fetal death. CC1 could have a particular role in this context, and such a result should be confirmed on a larger group of GBS isolated from *in utero* fetal death.

**KEYWORDS** *Streptococcus agalactiae*, CRISPR, typing, maternofetal infections, CC1, CC17, *in utero* death, maternal infection, molecular subtyping, neonatal infection

Address correspondence to Philippe Lanotte, philippe.lanotte@univ-tours.fr.

The authors declare no conflict of interest.

*S*treptococcus agalactiae, known as group B *Streptococcus* (GBS), is a major pathogen in humans. GBS emerged in humans in the 1960s and is now the leading bacterium responsible for maternal and fetal diseases (1, 2). Moreover, an increase in GBS infections in immunocompromised and elderly people has been observed since the late 1990s (3). GBS is usually a commensal bacterium of the vaginal and digestive tracts in

humans. The estimated carriage is about 15% to 30% in pregnant women, with differences according to world regions (4). Colonization during pregnancy can lead to neonatal disease in 1% to 2% of cases without any peripartum antibiotic prophylaxis (5–8). Two distinct syndromes of neonatal infections can occur. Early-onset disease (EOD) occurs during the first week of life following the inhalation or ingestion of vaginal secretions of GBS-colonized women during delivery. EOD is frequently characterized by pneumonia associated or not with bacteriemia and can sometimes lead to meningitis. Late-onset disease (LOD) occurs between 7 days and 3 months of life and is mostly characterized by bacteriemia, meningitis, or other invasive infections (arthritis, osteomyelitis), which are more frequent in LOD than in EOD (9, 10).

Capsular serotyping has contributed significantly to GBS descriptive epidemiology and is still relevant. Ten serotypes have been described in accordance with strain capsular polysaccharide composition, Ia, Ib, and II to IX (11). Serotype III strains were rapidly identified as responsible for the majority of neonatal diseases, especially in meningitis and LOD (12). More discriminating typing methods, making it possible to distinguish between isolates within the same serotype, were then developed. Multilocus sequence typing (MLST) is currently considered the reference technique for typing strains and investigating the population structure of GBS. MLST was originally developed for pathogenic microorganisms such as *Streptococcus pneumoniae* (13) and *Neisseria meningitidis* (14) and was then successfully adapted to *S. agalactiae* (15). The sequence type (ST) is determined by sequencing seven housekeeping genes and differs between strains according to their allelic profile. Closely related STs can be clustered in clonal complexes (CCs). This typing method has allowed the identification of a hypervirulent ST17 clone, which is particularly involved in neonatal diseases (15, 16).

The clustered regularly interspaced short palindromic repeats (CRISPR) and CRISPR-associated sequences (Cas) (CRISPR-Cas) system is an adaptive immune system protecting bacteria against invasive elements such as bacteriophages and plasmids, widely distributed in different bacterial clades, including in *S. agalactiae* (17). Newly acquired spacers from recent mobile genetic element (MGE) exposure are located at the leader end of the locus, while those found at the trailer end are ancestral spacers, which are conserved in phylogenetically related strains (18, 19). These phylogenetic relationships are correlated with the phylogenetic groups defined by MLST, allowing the use of CRISPR typing as an efficient alternative method to assess the CC of a strain. CRISPR sequence analysis has already been used for strain typing and subtyping in different bacterial species (20–22), including *S. agalactiae* (23). Several studies have been performed to finally propose CRISPR1 typing as an alternative method for CC assignment in GBS. An initial study showed that the type II-A CRISPR1 system is ubiquitous within the GBS species, and it revealed the extreme diversity of spacer content due to multiple acquisitions, duplications, or deletions (19). Strains were grouped according to the trailer-end spacers, which have proven to perfectly match the MLST classification. Another study, performed on 126 isolates, revealed major distinctive features in the CRISPR1 locus, according to the phylogenetic lineages previously defined by MLST (18). This study included three GBS reference strains (NEM316, 2603V/R, and A909) whose genomes are publicly available (24–26) and 123 epidemiologically unrelated GBS strains of human origin. These strains were previously characterized phenotypically and genotypically, determining their capsular serotype, ST, and prophage DNA content (27, 28). Two other studies, performed on 205 isolates recovered from 100 women (29) and 970 isolates from 10 GBS-colonized women, respectively (30), confirmed the ability of CRISPR1 typing to assign the correct CC compared to MLST. Finally, Beauruelle et al. explored the relevance of the CRISPR-based genotyping tool for GBS typing and compared it to current molecular methods, including MLST, by analyzing 255 GBS isolates. The authors demonstrated that the analysis of leader-end spacers, which vary more widely between strains, may offer stronger discriminating power than MLST (31). According to these data and previous studies, CRISPR1 typing has been proposed as a reference method for GBS typing (31).

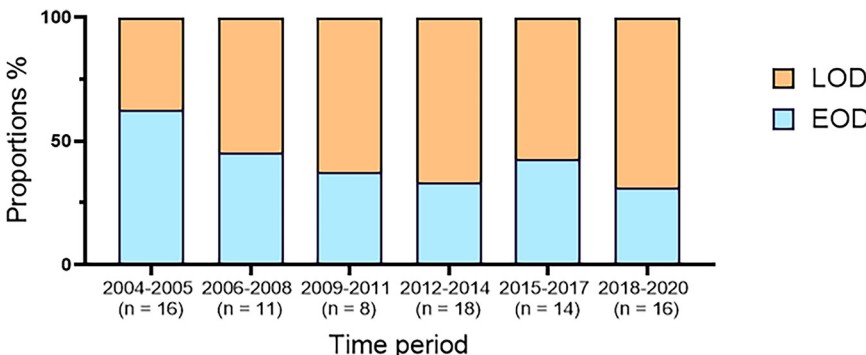

**FIG 1** Evolution of the proportion of EOD/LOD per time periods, from 2004 to 2020. This figure represents the proportions of EOD (blue) and LOD (orange) for each time period from 2004 to 2005 to 2018 to 2020. The total number of isolates for each time period is given below the *x* axis.

In this study, we used the CRISPR1 locus sequencing method to characterize GBS isolates responsible for maternofetal infectious disease at the University Hospital of Tours, France, between 2004 and 2020. All GBS isolates responsible for maternal or fetal infectious disease were analyzed, including GBS isolates from infections during pregnancy and responsible for *in utero* fetal death, which are generally less well studied.

## RESULTS

**Isolate distribution.** From 2004 to 2020, 115 GBS isolates were responsible for maternofetal infections in the University Hospital of Tours, France, and were analyzed in this work. Eighty-three of these isolates were isolated in a neonatal infection context, with 35 isolates responsible for EOD and 48 isolates responsible for LOD. Additionally, 32 isolates were isolated from a maternal infection during pregnancy or postpartum. Among the 32 isolates responsible for a maternal infection, 9 were isolated in the context of chorioamnionitis associated with *in utero* fetal death.

**Evolution of EOD and LOD.** Approximately five neonatal infections per year were diagnosed from 2004 to 2020, with an average of two EOD and three LOD per year. *In utero* fetal death remained rare. Isolates were grouped by 3-year periods, except for one 2-year period (2004 to 2005), to observe changes in EOD and LOD rates (Fig. 1). Over the 17-year period, the rate of EOD/LOD was reversed from about 2 EOD per 3 LOD in 2004 to 2005 to 2 LOD per 3 EOD in 2018 to 2020.

**CRISPR sequencing for GBS typing.** We analyzed the trailer end of the CRISPR1 locus containing the terminal direct repeat (TDR) and the ancestral acquired spacers of 115 isolates to characterize them and allow CC classification (supplemental files 1 and 2).

To identify CCs, we used the macro-enabled Excel tool providing known DR sequences and terminal DR sequences which are specific to CCs (Table 1). Indeed, this application helps extract CRISPR features from nucleotide sequences, removing the DR sequences and keeping only spacer sequences to finally represent them graphically in Excel spreadsheets. In order to determine the CC of isolates, only the sequences of ancestral spacers (one to four, going from the trailer end, according to the CC) and terminal direct repeats were considered and then compared to the dictionary of spacers established in previous studies and illustrated in supplemental file 2 (29–31). This method is correlated with MLST and has allowed the CC classification of 112 isolates. However, CRISPR1 analysis was not sufficiently discriminating between CC1 and CC19 for three isolates because they share the same ancestral spacers and TDR. We performed single nucleotide polymorphism (SNP) analysis on the 115 GBS isolates, which allows this discrimination (32). SNP analysis confirmed CC for all isolates except CC388, which cannot be identified by this method but is well distinguished using CRISPR analysis (supplemental file 1).

**Clonal complex distribution and evolution.** Among all 115 isolates, CC17 isolates were predominant, representing more than half of the isolates (60/115, 52.2%) (Fig. 2). Some isolates belonged to CC1 (18/115, 15,7%) and CC23 (14/115, 12.2%). CC10

**TABLE 1** Direct repeat sequences and terminal direct repeat sequences of the CRISPR1 locus according to clonal complexes

| Type | Repeat sequence (5′–3′) | Clonal complex(es) or ST |
|---|---|---|
| **Direct repeats** | | |
| Typical repeat | GTTTTAGAGCTGTGCTGTTTCGAATGGTTCCAAAAC | CC1, CC10, CC17, CC19, CC23 |
| Repeat variant | GTTTTAGAGCTGTGCTGTTTCAAATGGTTTCAAAAC | CC1, CC10, CC17, CC19, CC23 |
| | GTTTTAGACCTGTGCTGTTTCGAATGGTTCCAAAAC | CC1, CC10, CC17, CC19, CC23 |
| | GTTTTAGTGCTGTGCTGTTTCGAATGGTTCCAAAAC | CC17 |
| | GTTTTAGAGCTGTGCTATTTCGAATGGTTCCAAAAC | CC23 |
| | GTTTTAGAGCTGTGTTGTTTCGAATGGTTCCAAAAC | CC23 |
| | GTTTTAAAGCTGTGCTGTTTCGAATGGTTCCAAAAC | CC19 |
| | GTTTTAGAGCTGTGCTGTTTCGAATGATTCCAAAAC | CC22 |
| | | |
| **Terminal repeats** | | |
| Typical terminal repeat | GTTTTAGAGCTGTGCTGTTATTATGCTAGGACATCA | CC1, CC10, CC19 |
| Terminal repeat variants | GTTTTAAAGCTGTGCTGTTATTATGCTAGGGCACCA | CC23 |
| | GTTTTAGAGCTGTGCGGTTATTATGCTAGGGCACCG | CC17 |
| | GTTTTAGAGCTGTGCGGTTATTATGCTAGGGCATCA | CC1, CC10, CC19 |
| | GTTTTAGAGCTGCGCGGTTATTATGCTAGGGCATCA | CC10 (A909) |
| | GTTTTAGAGCTGTGCTGTTATTATGCTAGGGCACCA | CC388 |
| | GTTTTAGAGCTGCGCGGTTATTATGCTATGCTAGGA | CC22 |

isolates (10/115, 8.7%) and CC19 isolates (6/115, 5.2%) were less represented, and other CCs were extremely rare (CC22, 1 isolate; CC388, 3 isolates).

Interestingly, most isolates responsible for *in utero* fetal death associated with chorioamnionitis belonged to CC1 (6/9) (Fig. 3). Conversely, in maternal infections, the distribution of isolates was more diversified, with slightly more CC17 (7/23) and CC1 (5/23) isolates.

As expected, neonatal infections (EOD and LOD) were mainly represented by CC17 isolates (53/83, 63,9%). Other isolates belonged to CC23 (11/83, 13.3%), CC1 (8/83, 9.6%), CC10 (5/83, 6.0%), or CC19 (5/83, 6.0%). Also, one isolate belonged to CC388 (1.2%).

There was no significant change in the CC distribution of neonatal infection isolates over the period studied (data not shown).

**Clonal complex distribution of EOD and LOD.** Focusing on neonatal infections, LOD was almost fully represented by CC17 isolates (39/48, 81.3%). Few isolates belonged to other CCs (CC1 [3/48, 6.3%], CC19 [3/48, 6.3%], CC23 [2/48, 4.2%], or CC388 [1/48, 2.1%]). EOD was represented by a more homogeneous distribution of CCs, though it was still primarily represented by CC17 isolates (14/35, 40.0%). Interestingly, more CC23 isolates (9/35,

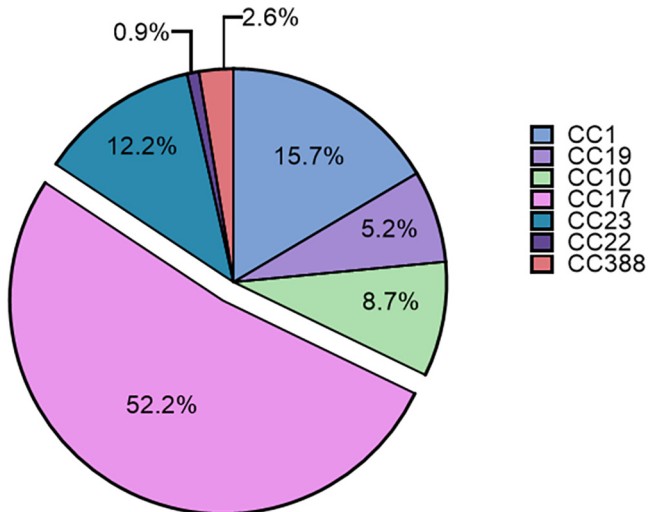

**FIG 2** Distribution of all isolates into clonal complexes. This figure represents the proportion of isolates into each clonal complex considering all the isolates of the study (*n* = 115).

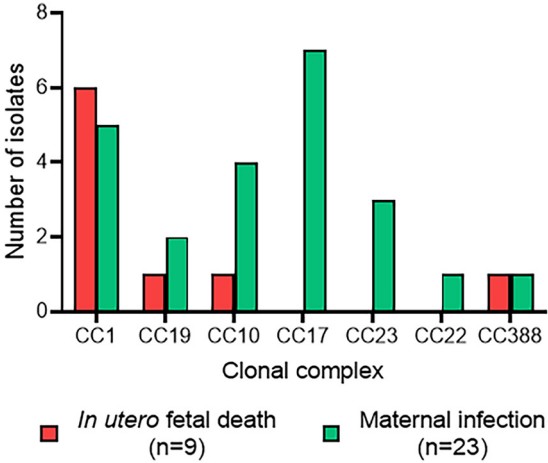

**FIG 3** Number of isolates responsible for *in utero* fetal death or maternal infection according to clonal complexes. The number of isolates responsible for *in utero* fetal death (red) and maternal infection (green) is shown for each clonal complex identified.

25.7%) were found in EOD than in LOD. CC10 isolates (5/35, 14.3%) were also only found in EOD (Fig. 4).

## DISCUSSION

With this work, we performed a descriptive analysis of GBS isolates responsible for maternal and fetal infections at the University Hospital of Tours from 2004 to 2020. Thus, CC17 was found to be predominant in overall isolates and especially among isolates from late-onset neonatal diseases as already shown in several studies (15, 16, 33). Unexpectedly, most cases of *in utero* fetal death involved isolates belonging to CC1.

We used CRISPR1 locus analysis, an alternative typing method as relevant as MLST, to type our 115 isolates (15). This method has already been used successfully in several studies (19), including studies from our laboratory (18, 29–31). CRISPR1 locus analysis allows the classification of GBS isolates into phylogenetic clusters, which are highly correlated with CCs determined by MLST, thanks to the specific sequences of the TDR and ancestral spacers. Analyzing the ancestral spacers and TDR composition of the CRISPR locus is convenient, easy to perform, and less time-consuming than MLST. Indeed, by single-locus sequencing, this method offers a good compromise between discriminatory power, phylogenetic data, and simplicity compared to the reference method. Thus, this method recently replaced MLST in our hospital laboratory to type GBS strains

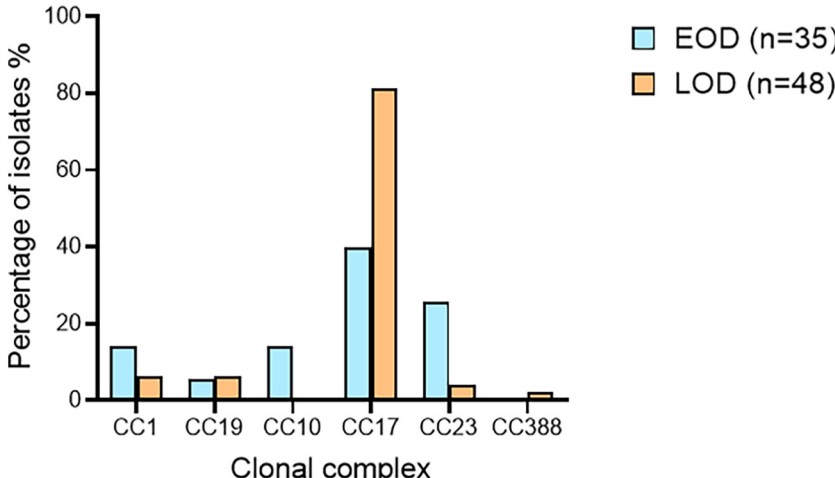

**FIG 4** Distribution of EOD and LOD isolates according to clonal complexes.

responsible for invasive infections. Furthermore, spacers acquired more recently can distinguish between different isolates within a given CC, which increases the discriminatory power of CRISPR1 typing compared to MLST (31). One of the limitations of the CRISPR1 typing technique is the difficulty in distinguishing between CC1 and CC19, which share the same ancestral spacer and the same TDR. This difficulty could be overcome by analyzing more spacers than for other isolates (third, fourth, or more, going from the trailer end) or by using a complementary method if no acquisition or a deletion occurs in the strain. CRISPR1 typing was completed by SNP determination for three isolates to attribute them to the correct CC. Actually, a BLAST analysis of the full CRISPR1 sequence obtained could have been done to determine the sequence type of these isolates.

Our study presents some limitations. First, this study was monocentric: all isolates were recovered and analyzed at the University Hospital of Tours, France. This presents the advantage of showing the local epidemiological characteristics of GBS invasive isolates in maternal and neonatal infections. However, a study performed in multiple centers would have increased the number of isolates, bringing more data, especially regarding the subgroups of isolates associated with *in utero* fetal death. Moreover, it could be interesting to complete our work using another method such as whole-genome sequencing (WGS) to confirm our findings and obtain more data. Performing WGS on isolates associated with *in utero* fetal death, for example, could identify interesting features such as specific virulence factors.

A CC was attributed to all 115 GBS isolates responsible for a maternofetal infection between 2004 and 2020 using the CRISPR1 typing method for the representative strains A909, BM110, NEM316, and 2603V/R. Our collection was mainly represented by CC1, CC10, CC17, CC19, and CC23 isolates, which are the most prevalent CCs in humans, whether in infection or colonization (15, 16). Among neonatal infection isolates, CC17 was predominant, especially among LOD isolates, as also reported by the French National Reference Centre for streptococcal infections (NRC) (10). The link between CC17 and neonatal diseases has long been demonstrated in several countries (15, 16, 33, 34). Indeed, these isolates are responsible for 80% of the most severe cases of GBS infection, including meningitis, because of specific virulence factors previously described (10, 35–37).

In addition to specific virulence factors, CC17 strains also show notable differences in their CRISPR1 locus, which contains significantly fewer spacers than other lineages (18). A deletion of the regular *cas* promoter in ST17 strains, without preventing the functionality of the system, was also demonstrated (38). These studies on the GBS CRISPR-Cas system have deepened our understanding of its functioning and the particularities found in hypervirulent strains. Moreover, the CRISPR-Cas system could potentially be involved in the virulence of CC17 strains in some way, though the regulatory pathways and their role remain to be demonstrated (39).

In our collection, isolates distribution showed a decrease in EOD over time. Such a decrease has already been described in several countries following the publication of guidelines for the prevention of perinatal group B streptococcal disease (1, 40, 41). However, the incidence of GBS LOD has remained unchanged since the 20th century. As a reminder, LODs are almost exclusively represented by CC17 strains that present particular virulence factors and differences in their CRISPR1 locus. Considering that the screening methods currently used are not efficient in preventing LOD, developing new screening techniques based on a virulence factor or on CRISPR1 locus specificity of CC17 strains could be an alternative worth considering. Using such tools, early detection of CC17 strains in pregnant women, mothers, or newborns before a declared infection may help to reduce the incidence of LOD over time, using prophylactic treatments in these cases. Unfortunately, screening methods will still lack sensitivity to avoid any LOD because GBS carriage is known to be intermittent. Nevertheless, the search for more efficient diagnostic tools is essential, and using CRISPR-Cas systems may be an interesting way to achieve this. An assay using *in vitro* CRISPR-Cas13 detection of

specific nucleic acids was recently developed in GBS screening and showed enhanced performance over culture- and PCR-based assays (42).

Surprisingly, the few isolates responsible for *in utero* fetal death in our study were mainly part of CC1, and no CC17 was found. A predominance of CC1 isolates expressing capsular serotype VI was shown in nonpregnant adult infections in Taiwan, but only a few CC1 infections occurred in newborns, and there were no data concerning *in utero* fetal death (43). Recently, Schindler et al. also reported a possible association between ST1 and intrauterine fetal death (44). These results highlight a potential role of CC1 in *in utero* disease. Actually, GBS maternofetal infections are often summarized as EOD and LOD, but an invasive infection earlier during pregnancy can occur and lead to preterm birth, stillbirth, or fetal death (45). Nearly 57,000 fetal infections or stillbirths were attributed to GBS in 2015 according to an analysis of international data (46). However, in most studies, molecular characteristics, serotype, and CC of isolates from *in utero* fetal death were not shown or sometimes associated with EOD due to a low number of cases. Our findings suggest that isolates from *in utero* fetal death may differ from those responsible for EOD or LOD, and this should be investigated in a larger population.

In conclusion, we described the local distribution of GBS isolates responsible for maternal and neonatal diseases. We used a CRISPR typing method, which is correlated with MLST but faster and easier to perform. CC17 isolates expressing specific virulence factors were found to be predominant in neonatal diseases, particularly in LOD. On the other hand, CC1 isolates predominate in infections responsible for *in utero* fetal death, which was rarely reported to date. More epidemiological data are needed, but specifically exploring the characteristics of these clones should be of interest.

## MATERIALS AND METHODS

**Bacterial isolates.** One hundred fifteen GBS isolates responsible for a maternofetal infection were isolated at the University Hospital of Tours, France, between 2004 and 2020. These isolates represent all *S. agalactiae* recovered in blood culture (*n* = 77), cerebrospinal fluid (*n* = 12), and other sterile sites such as joint fluid (*n* = 3), breast abscess (*n* = 8), and placenta (*n* = 15) sampled from newborns (*n* = 83), aborted or stillborn fetus (*n* = 9), and pregnant women (*n* = 23). All isolates were stored at −80°C before being analyzed by the method described below. Moreover, *S. agalactiae* NEM316, 2603V/R, BM110, and A909 were selected as representative strains of the ST23 (CC23), ST110 (CC19), ST17 (CC17). and ST7 (CC10) lineages, respectively. These strains are well characterized, including by genome sequencing, MLST typing, and CRISPR1 typing, and were isolated from human invasive infections (18, 24–26). CRISPR1 typing was performed once more on these strains to validate the results.

**DNA extraction.** DNA extraction was performed with a heating method. One colony of bacteria was added to a PCR amplification mix, and the first heating step was sufficient to extract GBS DNA as previously used (47).

**CRISPR1 locus amplification and sequencing.** CRISPR1 locus amplification was performed in a T3000 thermocycler (Biometra) using CRISPR1-PCR-F and CRISPR1-PCR-R primers targeting flanking regions of the CRISPR1 locus as previously described (18, 19, 29). Briefly, 0.5 $\mu$M each primer, 0.2 mM deoxynucleoside triphosphate (dNTP), 2 mM MgCl$_2$, 0.02 U/$\mu$L GoTaq polymerase (Promega), 1× PCR buffer, and a small GBS colony were mixed in a total volume of 25 $\mu$L. The PCR mixtures were heated to 94°C for 5 min, followed by 40 cycles of a denaturation step at 94°C for 30 s, an annealing step at 55°C for 30 s, and an elongation step at 72°C for 1 min, ending with a final extension step at 72°C for 7 min. PCR amplification was verified by electrophoretic migration in a 1% agarose gel, and then the PCR products were purified using centrifugal filter units (Millipore Corporation) in accordance with the manufacturer's recommendations.

CRISPR1 locus sequencing from purified PCR products was then performed by the Sanger sequencing technique on an ABI 3500 Dx genetic analyzer (Applied Biosystems, Thermo Fisher Scientific), using CRISPR1-SEQ-F and CRISPR1-SEQ-R internal sequencing primers and BigDyeTerminator mix v3.1 (Applied Biosystems) as previously described (19).

**CRISPR1 analysis.** DNA sequences were analyzed using the ApE, A plasmid Editor, v2.0.47. This software was used to check the fluorescence signal of each base and remove the parts where bases were not well individualized and could lead to a miscalled base in the sequence. Both forward and reverse sequences were analyzed for each isolate. A macro-enabled Excel tool (P. Horvath, International Flavors and Fragrances [IFF]) was then used to identify spacers, direct repeats (DRs), and flanking regions and to make graphical illustrations of features of interest.

**SNP analysis.** This method is based on an allele-specific real-time PCR assay that explores four single nucleotide polymorphisms (SNPs) located in three of the seven housekeeping genes used in the MLST method. This method was carried out as previously described (32).

## SUPPLEMENTAL MATERIAL

Supplemental material is available online only.
**SUPPLEMENTAL FILE 1**, DOCX file, 0.1 MB.
**SUPPLEMENTAL FILE 2**, PDF file, 0.5 MB.

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
