## [Reviewer comments · Microbiology Spectrum]

Microbiology Spectrum

Group B Streptococcus CRISPR1 typing of maternal, fetal and neonatal infectious disease isolates highlights the importance of CC1 in in utero fetal death

Brice Le Gallou, Adeline Pastuszka, Coralie Lemaire, Laurent Mereghetti, and Philippe Lanotte

Corresponding Author(s): Philippe Lanotte, Universite de Tours

Review Timeline:

Submission Date:	December 22, 2022
Editorial Decision:	February 13, 2023
Revision Received:	April 18, 2023
Editorial Decision:	May 6, 2023
Revision Received:	May 23, 2023
Accepted:	May 24, 2023

Editor: Silvia Cardona

Reviewer(s): Disclosure of reviewer identity is with reference to reviewer comments included in decision letter(s). The following individuals involved in review of your submission have agreed to reveal their identity: Laura Maria Andrade de Oliveira (Reviewer #2)

Transaction Report:

DOI: <https://doi.org/10.1128/spectrum.05221-22>

February 13, 2023

Prof. Philippe Lanotte
Institut National de Recherche pour l'Agriculture l'Alimentation et l'Environnement Centre Val de Loire
UMR1282-ISP University of Tours -Bacteria and Maternofetal Risk
CHRU de Tours
Tours F-37044
France

Re: Spectrum05221-22 (Group B Streptococcus CRISPR1 typing of maternal, fetal and neonatal infectious disease isolates highlights the importance of CC1 in in utero fetal death)

Dear Prof. Philippe Lanotte:

Thank you for submitting your manuscript to Microbiology Spectrum. Your work has been reviewed by two experts in the field. Both reviewers noted that your work has merit. However, reviewers also noted that there are significant shortcomings that prevent the manuscript to be accepted in its current form. You will find the reviewers' comments and instructions below.

Please, pay special attention to the following:

CRISPR1 typing is applied to assign the GBS isolates to clonal complexes defined by MLST, but your work do not show the GBS isolates with the conventional MLST analysis to validate their results. Please, note that your work will not be accepted if you don't provide experimental data to address this comment.

Please, review the methodological design of CRISPR1 typing, and results and ensure the conclusions are supported by the data.

In addition, please consider improving the clarity of the figures. Figures are not only added to provide evidence of the work but to facilitate understanding. Visually attractive figures can have a positive effect during the peer review. To improve clarity, I recommend the manuscript to be critically read by a non-author who is not an expert in the field.

Link Not Available

Sincerely,

Silvia Cardona

Journals Department
Reviewer comments:

Reviewer #1 (Comments for the Author):

Summary:

The authors analyzed 115 GBS isolates with CRISPR typing. They found CC17 to be predominant overall and among isolates with late onset disease. They found several CC1 isolates among isolates with fetal death in utero. This is an interesting manuscript that defines the GBS population locally. The method used is not standardized and involves sequencing of individual loci and correlation with ST, but not actual ST determination by MLST or WGS.

General comments:

Include the main findings and conclusions at the beginning and end of the discussion section.

Explain why CRISPR typing was used instead of other typing methods and include the basic strengths and limitations of using this approach.

How do your results differ from other results?

Explain how the results were quality assured and whether QC strains were used for the study.

Include the limitations sections because almost all studies have at least some of these.

Minor comments:

Several abbreviations in the text are not explained: DR, , DRT, MGE etc.

Reviewer #2 (Comments for the Author):

In this study, Brice and colleagues used the CRISPR1 locus sequencing method to characterize GBS isolates responsible for maternofetal infectious disease at the University Hospital of Tours, France, between 2004 and 2020. The authors assigned the GBS isolates to five different clonal complexes, mainly CC17 and CC1, by the CRISPR1 typing method. CC1 was the most common among GBS isolates associated with in utero death and the authors highlights the possibility of a particular role of this clonal complex in in-utero infection. Lastly, the authors suggest that isolates from in utero fetal death may differ from those responsible for EOD or LOD, and this should be investigated on a larger population. This is an interesting research article with clear goals and study parameters and procedures. However, there are some considerations to be addressed by the authors, as indicated below.

Overall, the manuscript should be revised regarding the language and the quality of figures should be improved. In addition, the manuscript should be revised regarding the CRISPR1 typing results and the corresponding conclusions shown by the authors in this paper. Although the experimental design of the study is acceptable, the manuscript does not clearly present robust and accurate data regarding CRISPR1 typing.

Abstract:

Although the title of the manuscript highlights the importance of CRISPR1 typing of GBS strains associated with neonatal disease, CRISPR typing results are not addressed properly in the abstract. The authors include prevalence data of EOD and LOD overtime and the clonal complexes associated with neonatal disease cases, but at any moment the CRISPR1 typing method is cited or discussed.

Lines 106-108: please provide the number of GBS isolates recovered from each clinical source (blood culture, cerebrospinal fluid, or other sterile sites such as joint fluid or placenta) and origin (newborns, aborted or stillborn fetus, or pregnant women).

Lines 120-121: sequencing of PCR products was performed by Sanger sequencing technique? By using which equipment? Please include these methodological details in the text.

Lines 123-127: What was the rationale behind the analysis of CRISPR sequences? Which CRISPR features were considered in this analysis? Which dictionary of spacers was used? Please include in the text all these methodological details. In addition, the authors should describe in more detail the macro-enabled Excel tool used in the analysis of CRISPR sequences and provide a copy of the spreadsheet as a supplementary material.

Lines 159-161: Were the GBS isolates previously submitted to the MLST conventional analysis (sequencing the 7 housekeeping genes of the GBS scheme)? This result is crucial to validate the CRISPR1 typing analysis to assign clonal complexes shown by

the authors in this study. The results of CRISPR1 typing method performed in this study must be compared to the results of CC assignment by the conventional MLST analysis for the GBS isolates analyzed in the study to validate the results obtained.

Lines 169-170: The results of single nucleotide polymorphism analysis of GBS isolates not distinguished by CRISPR1 typing method must be shown in the paper.

Staff Comments:

Preparing Revision Guidelines

Please return the manuscript within 60 days; if you cannot complete the modification within this time period, please contact me. If you do not wish to modify the manuscript and prefer to submit it to another journal, please notify me of your decision immediately so that the manuscript may be formally withdrawn from consideration by Microbiology Spectrum.

31 January 2023

**Editor
Spectrum**

Dear **Editor**,

I have thoroughly read the manuscript entitled: "*Group B Streptococcus CRISPR1 typing of maternal, fetal and neonatal infectious disease isolates highlights the importance of CC1 in in utero fetal death*", and have given it my full consideration.

Summary:

The authors analyzed 115 GBS isolates with CRISPR typing. They found CC17 to be predominant overall and among isolates with late onset disease. They found several CC1 isolates among isolates with fetal death in utero. This is an interesting manuscript that defines the GBS population locally. The method used is not standardized and involves sequencing of individual loci and correlation with ST, but not actual ST determination by MLST or WGS.

General comments:

Include the main findings and conclusions at the beginning and end of the discussion section. Explain why CRISPR typing was used instead of other typing methods and include the basic strengths and limitations of using this approach.

How do your results differ from other results?

Explain how the results were quality assured and whether QC strains were used for the study.

Include the limitations sections because almost all studies have at least some of these.

Minor comments:

Several abbreviations in the text are not explained: DR, , DRT, MGE etc.

Strengths:

Disclosing local epidemiology of GBS in all major disease types

Limitations:

Poor structure of the discussion

Niche method used for typing

Conclusion:

Modifications

Reviewer #1 (Comments for the Author):

Summary:

The authors analyzed 115 GBS isolates with CRISPR typing. They found CC17 to be predominant overall and among isolates with late onset disease. They found several CC1 isolates among isolates with fetal death in utero. This is an interesting manuscript that defines the GBS population locally. The method used is not standardized and involves sequencing of individual loci and correlation with ST, but not actual ST determination by MLST or WGS.

General comments:

1) Include the main findings and conclusions at the beginning and end of the discussion section.

We followed the reviewer's advice and included the main findings and conclusions at the beginning (lines 227-230 of the revised manuscript) and end of the discussion section (lines 315-322).

2) Explain why CRISPR typing was used instead of other typing methods and include the basic strengths and limitations of using this approach.

MLST is the reference typing method for GBS nowadays, but remains time-consuming and generates large volumes of data (7 sequences of nearly 500bp per isolate). With other teams, we participated in the development of the CRISPR typing method which presents a well-demonstrated discriminatory power and defines groups that perfectly match MLST-based clustering. Using single locus sequencing, this method offers a good compromise between discriminatory power, phylogenetic data and simplicity (Lopez-Sanchez et al., 2012, Lier et al., 2015, Beauruelle et al., 2017; Beauruelle et al., 2018, Beauruelle et al., 2021). We modified the text in order to explain more precisely why CRISPR typing was used, including the basic strengths and limitations of this method (lines 236-248). Nevertheless, CRISPR typing is quite less discriminant for CC1/CC19 isolates. Thus we have determined SNP profile, a method avoiding a sequencing step, to complete typing. Results are presented extensively as supplementary data.

3) How do your results differ from other results?

Firstly, the aim of this study was to provide data regarding the local epidemiology (University hospital of Tours) of GBS strains responsible for invasive infections in mothers and newborns, including infections associated with *in utero* fetal death. The main clonal complex represented in newborn infections, especially in late-onset disease, was CC17, which was already known and described in several studies. Our data do not differ from other results regarding this conclusion, but we were able to show it using the CRISPR typing method which has rarely been used.

We were also able to show some interesting results concerning isolates responsible for infections associated with *in utero* fetal death. The main CC implicated was unexpectedly shown to be CC1, without any CC17 isolates, which was only reported in a recent Israeli study (Schindler *et al.* 2020) (lines 304-305). Actually, we had few isolates to analyze in our study, and this is why we suggest performing a typing study on a larger number of isolates responsible for *in utero* fetal death. This result is the main finding that we wanted to highlight, because CC1 is more frequently isolated in non-pregnant adult infections (Tsai *et al.* 2019) and not specifically known to be implicated in neonatal or *in utero* infections. This last reference was added in the revised version of the manuscript.

4) Explain how the results were quality assured and whether QC strains were used for the study.

We assured the quality of our results in two steps. First, as specified in the text, the PCR amplification was verified by electrophoretic migration in a 1% agarose gel (line 144-146). Then, concerning Sanger

sequencing data, we used the ApE- A plasmid Editor v2.0.47© for analysis of sequences. The first and last parts of the sequence obtained were removed, and only the well-characterized sequence on the chromatogram with unique peaks was used for each isolate (lines 153-159).

During previous studies, four well-characterized GBS strains: NEM316, A909, BM110 and 2603V/R (Glaser *et al.*, 2002; Tettelin *et al.*, 2002; 2005), were analyzed by this method (Lier *et al.*, 2015). These strains were considered as QC strains and we repeated the analysis to validate our current results (lines 125-130).

Moreover, DNA sequences of the terminal direct repeat (TDR) and the spacers expected for each major clonal complex are known and have been published (Beauruelle *et al.*, 2021). Their graphic representation is shown in the table 2.

5) Include the limitations sections because almost all studies have at least some of these.

We added a paragraph on limitations to the discussion section. The main ones are the monocentric aspect of the study, and the low number of isolates associated with *in utero* fetal death (lines 252-261).

Minor comments:

Several abbreviations in the text are not explained: DR, , DRT, MGE etc.

We corrected this oversight and took care to explain all abbreviations used in the text.

Reviewer #2 (Comments for the Author):

In this study, Brice and colleagues used the CRISPR1 locus sequencing method to characterize GBS isolates responsible for maternofetal infectious disease at the University Hospital of Tours, France, between 2004 and 2020. The authors assigned the GBS isolates to five different clonal complexes, mainly CC17 and CC1, by the CRISPR1 typing method. CC1 was the most common among GBS isolates associated with in utero death and the authors highlights the possibility of a particular role of this clonal complex in in-utero infection. Lastly, the authors suggest that isolates from in utero fetal death may differ from those responsible for EOD or LOD, and this should be investigated on a larger population. This is an interesting research article with clear goals and study parameters and procedures. However, there are some considerations to be addressed by the authors, as indicated below.

1) Overall, the manuscript should be revised regarding the language and the quality of figures should be improved.

The language was revised by asking a specialized organization, independent of our team, and we worked on improving the quality of figures.

2) In addition, the manuscript should be revised regarding the CRISPR1 typing results and the corresponding conclusions shown by the authors in this paper.

We revised our manuscript regarding the CRISPR1 typing results and the conclusions we made. Details are shown below.

3) Although the experimental design of the study is acceptable, the manuscript does not clearly present robust and accurate data regarding CRISPR1 typing.

We revised our manuscript in order to explain more precisely the robustness of CRISPR1 typing method. Details are shown below.

Abstract:

Although the title of the manuscript highlights the importance of CRISPR1 typing of GBS strains associated with neonatal disease, CRISPR typing results are not addressed properly in the abstract. The authors include prevalence data of EOD and LOD overtime and the clonal complexes associated with neonatal disease cases, but at any moment the CRISPR1 typing method is cited or discussed.

We changed the abstract to provide further information concerning the use of the CRISPR1 typing method to obtain the CC affiliation of our isolates. (Lines 19-24).

Lines 106-108: please provide the number of GBS isolates recovered from each clinical source (blood culture, cerebrospinal fluid, or other sterile sites such as joint fluid or placenta) and origin (newborns, aborted or stillborn fetus, or pregnant women).

We added the number of GBS isolates recovered from each clinical source and origin (lines 121-123). Also, a supplementary data table summarizing all data of each isolate (origin, source, date of isolation, SNP result and CRISPR typing result with spacer representations) has been added. Additionally, when working on the supplemental data table, a transcription error on one isolate was observed and fixed, and we apologize for that. This error had no impact on the conclusions we made (line 203 and figures 2 and 3).

Lines 120-121: sequencing of PCR products was performed by Sanger sequencing technique? By using which equipment? Please include these methodological details in the text.

We included these methodological details in the text (lines 147-150). We also provided details concerning the amplification protocol (lines 136-146)

Lines 123-127: What was the rationale behind the analysis of CRISPR sequences?

As we have specified in the text (lines 83-88), CRISPR-Cas systems, by acquiring a part of DNA from MGE, reflect the different encounters between a bacterium and its environment. Thus, analysis of these different recent encounters can provide more discriminant power between strains even within a single clonal complex than MLST. On the other hand, ancestral spacers are conserved within clonal complexes defined by MLST, which explains the possibility of using this sequence analysis in epidemiological studies as an alternative to MLST.

Which CRISPR features were considered in this analysis?

Even if we were able to sequence the whole locus, only the terminal direct repeat and terminal spacers (one to four according to the clonal complex; table 2) were considered in this analysis, in order to group isolates into clonal complexes related to the MLST method. We have covered this in greater detail in the text (lines 185-195 and 236-241).

Which dictionary of spacers was used?

We used the dictionary of spacers made in the previous studies and published by Beauruelle *et al.*, 2021.

Please include in the text all these methodological details.

In addition, the authors should describe in more detail the macro-enabled Excel tool used in the analysis of CRISPR sequences and provide a copy of the spreadsheet as a supplementary material.

We have added further details of the tools used (lines 153-159) and the cleaned sequences, containing only spacers as graphic representations were added to the supplementary data.

Lines 159-161: Were the GBS isolates previously submitted to the MLST conventional analysis (sequencing the 7 housekeeping genes of the GBS scheme)? This result is crucial to validate the CRISPR1 typing analysis to assign clonal complexes shown by the authors in this study. The results of CRISPR1 typing method performed in this study must be compared to the results of CC assignment by the conventional MLST analysis for the GBS isolates analyzed in the study to validate the results obtained.

The GBS isolates analyzed by CRISPR1 typing in this work were not analyzed by the MLST method. Indeed, sequential studies had already been performed to validate the CRISPR1 typing as an alternative method for CC assignment. We added more details about this to the introduction section (lines 88-110). Over the years, this alternative method has provided many data correlated to MLST. Thus, for the present study, we considered CRISPR1 typing to be validated, and our objective was to use it instead of MLST, because of its conveniency, while also expanding the library of spacers.

Lines 169-170: The results of single nucleotide polymorphism analysis of GBS isolates not distinguished by CRISPR1 typing method must be shown in the paper.

We added the results of single nucleotide polymorphism analysis to supplementary data, which summarizes all isolate-related data (origin, source, date of isolation, SNP result and CRISPR typing result with spacer representations).

May 6, 2023

Prof. Philippe Lanotte
Universite de Tours
UMR1282-ISP University of Tours -Bacteria and Maternofetal Risk
CHRU de Tours
Tours F-37044
France

Re: Spectrum05221-22R1 (Group B Streptococcus CRISPR1 typing of maternal, fetal and neonatal infectious disease isolates highlights the importance of CC1 in in utero fetal death)

Dear Prof. Philippe Lanotte:

Thank you for submitting your manuscript to Microbiology Spectrum. As you will see your paper is very close to acceptance. Please modify the manuscript along the lines recommended by a reviewer. As these revisions are quite minor, I expect that you should be able to turn in the revised paper in less than 30 days, if not sooner.

When submitting the revised version of your paper, please provide (1) point-by-point responses to the issues raised by the reviewers as file type "Response to Reviewers," not in your cover letter, and (2) a PDF file that indicates the changes from the original submission (by highlighting or underlining the changes) as file type "Marked Up Manuscript - For Review Only". Please use this link to submit your revised manuscript. Detailed instructions on submitting your revised paper are below.

Link Not Available

Sincerely,

Silvia Cardona

Reviewer comments:

Reviewer #1 (Comments for the Author):

Much improved manuscript. Thank you for your explanations.

Reviewer #2 (Comments for the Author):

The manuscript was extensively revised, and the authors addressed all comments accordingly, although there is a last suggestion to be made. Table 2 should include a legend for each graphical representation of each spacer and each DRT.

Preparing Revision Guidelines

To submit your modified manuscript, log onto the eJP submission site at <https://spectrum.msubmit.net/cgi-bin/main.plex>. Go to Author Tasks and click the appropriate manuscript title to begin the revision process. The information that you entered when you

first submitted the paper will be displayed. Please update the information as necessary. Here are a few examples of required updates that authors must address:

Please return the manuscript within 60 days; if you cannot complete the modification within this time period, please contact me. If you do not wish to modify the manuscript and prefer to submit it to another journal, please notify me of your decision immediately so that the manuscript may be formally withdrawn from consideration by Microbiology Spectrum.

Response to reviewers :

Reviewer #1 (Comments for the Author):

No comment

Reviewer #2 (Comments for the Author):

The manuscript was extensively revised, and the authors addressed all comments accordingly, although there is a last suggestion to be made. Table 2 should include a legend for each graphical representation of each spacer and each DRT.

We have transferred table 2 to Supplemental table 1 to answer to this point. In fact with a detailed legend, it seems to us that these information's could find a better place as supplementary table. Due to this modification, Supplemental table S1 become Supplemental table S2. We have modified the manuscript to introduce these modifications on lines 192 and 199.

May 24, 2023

Prof. Philippe Lanotte
Universite de Tours
UMR1282-ISP University of Tours -Bacteria and Maternofetal Risk
CHRU de Tours
Tours F-37044
France

Re: Spectrum05221-22R2 (Group B Streptococcus CRISPR1 typing of maternal, fetal and neonatal infectious disease isolates highlights the importance of CC1 in in utero fetal death)

Dear Prof. Philippe Lanotte:

Your manuscript has been accepted, and I am forwarding it to the ASM Journals Department for publication. You will be notified when your proofs are ready to be viewed.

Sincerely,

Silvia Cardona
Editor, Microbiology Spectrum
